# Efficient and Stable Coupling of the SuperdropNet Deep Learning-based Cloud Microphysics (v0.1.0) to the ICON Climate and Weather Model (v2.6.5)

Caroline Arnold[1,2,3], Shivani Sharma[2,3,4], Tobias Weigel[1,2,3], and David Greenberg[2,3]

[1]German Climate Computing Center DKRZ, Hamburg, Germany
[2]Helmholtz Zentrum Hereon, Geesthacht, Germany
[3]Helmholtz AI, Munich, Germany
[4]International Max-Planck-Research School on Earth System Modeling, Hamburg, Germany

**Correspondence:** Caroline Arnold (arnold@dkrz.de), Shivani Sharma (shivani.sharma@hereon.de)

**Abstract.**

Machine learning (ML) algorithms can be used in Earth System models (ESMs) to emulate sub-grid-scale processes. Due to the statistical nature of ML algorithms and the high complexity of ESMs, these hybrid ML-ESMs require careful validation. Simulation stability needs to be monitored in fully coupled simulations, and the plausibility of results needs to be evaluated in suitable experiments.

We present the coupling of SuperdropNet, a machine learning model for emulating warm rain processes in cloud microphysics, into ICON (Icosahedral Nonhydrostatic) 2.6.5. SuperdropNet is trained on computationally expensive droplet based simulations and can serve as an inexpensive proxy within weather prediction models. SuperdropNet emulates the collision-coalescence of rain and cloud droplets in a warm rain scenario and replaces the collision-coalescence process in the two-moment cloud microphysics scheme. We address the technical challenge of integrating SuperdropNet, developed in Python and PyTorch, into ICON, written in Fortran, by implementing three different coupling strategies: embedded Python via the C Foreign Function Interface, pipes, and coupling of program components via YetAnotherCoupler (YAC). We validate the emulator in the warm bubble scenario and find that SuperdropNet runs stable within the experiment. In comparing experiment outcomes from the bulk moment scheme and SuperdropNet, we find that the results are physically consistent, and discuss differences that are observed for several diagnostic variables.

In addition, we provide a quantitative and qualitative computational benchmark for three different coupling strategies—embedded Python, coupler YAC, and pipes—and find that embedded Python is a useful software tool for validating hybrid ML-ESMs.

## 1 Introduction

Machine learning (ML) is increasingly used in Earth system models (ESMs) to emulate sub-grid-scale processes that are typically parameterized or neglected due to their high computational cost (Christensen and Zanna, 2022; Dueben et al., 2021; Irrgang et al., 2021; Gentine et al., 2018). ML algorithms are statistical algorithms that are trained on data. Neural networks

are a widely used class of ML algorithms. They contain trainable parameters, the weights and biases, that are learned from data by minimizing a cost function. The trained algorithm can then be used for inference, i.e. application on unseen data of the same kind. When sub-grid-scale processes are replaced by ML algorithms, the improvement can aim at speeding up the overall simulation by emulating the existing parameterization. This was first established by using neural networks to emulate long-wave radiative transfer (Chevallier et al., 2000; Krasnopolsky et al., 2005). Recent examples include the emulation of gravitational wave drag (Chantry et al., 2021), cloud microphysics (Brenowitz et al., 2022), the ocean in a coupled climate model (Sonnewald et al., 2021), and cloud radiative effects (Meyer et al., 2022).

Other studies aim to improve the overall description of the Earth system by providing a better parameterization. ML algorithms can be trained on high-resolution ESM output or even on separately simulated processes to emulate resolved processes in a low-resolution simulation, e.g. for gravity waves (Dong et al., 2023), cloud cover parameterizations (Grundner et al., 2022), general parameterizations (Brenowitz and Bretherton, 2018), sub-grid-scale momentum transport (Yuval and O'Gorman, 2023), effects of cloud resolving simulations (Rasp et al., 2018), ozone distributions (Nowack et al., 2018), and radiative transfer (Belochitski and Krasnopolsky, 2021).

Many parameterizations in ESMs can be removed at higher resolutions if the process can be completely resolved, such as the convective parameterizations. On the other hand, some others would need to be parameterized even for 1-km scale weather models. Cloud microphysical processes fall in this category. Processes dealing with the droplet interactions that lead to precipitation are lumped together and referred to as cloud microphysical processes. Due to high particle counts even at small grid sizes and our incomplete understanding of processes that occur at a molecular level in clouds (Morrison et al., 2020), we cannot expect cloud microphysical parameterizations to become obsolete in the near future for high resolution models.

The parameterization of these processes suffers from a unique accuracy/speed trade-off. The most accurate droplet based Lagrangian schemes such as the superdroplet method (Shima et al., 2009) are computationally expensive. The commonly used bulk moment schemes represent the complex particle size distributions as only the first two moments, referring to the total droplet concentration and the total water content of the hydrometeors. For modelling the droplet collisions in a warm-rain scenario, ICON uses the well studied bulk moment scheme developed in Seifert and Beheng (2001). To bridge this gap and to make the use of more complex microphysical schemes feasible within operational models, a data-driven approach can be employed. We present here the integration of SuperdropNet (Sharma and Greenberg, 2024), an ML algorithm for emulating warm rain processes in cloud microphysics, into ICON 2.6.5. SuperdropNet is trained on zero dimensional box model superdroplet simulations from McSnow 1.1.0 (Brdar and Seifert, 2018), a superdroplet based cloud microphysics model, in a warm rain scenario and replaces the warm rain processes in the two-moment scheme available in ICON 2.6.5 (Seifert and Beheng, 2006).

Due to the statistical nature of ML algorithms and the complex nonlinear interactions in ESMs, hybrid systems of numerical ESMs and ML algorithms require careful validation and verification (Dueben et al., 2022; Brenowitz and Bretherton, 2019). Stand-alone ML algorithms are first trained on a dataset and then validated on a hold-out test dataset that is not seen during training. This test set is within the distribution of the training data. When an ML algorithm is coupled to an ESM, it may encounter conditions outside of the range of the training data, and the required extrapolation could lead to instabilities

(Yuval et al., 2021). Thus, the so-called *offline* performance of an ML algorithm is often not a good indicator of its *online* performance (Brenowitz et al., 2020b; Rasp, 2020). Stability is a major concern when introducing ML emulators into ESMs. It can be improved by adapting the training procedure (Qu and Shi, 2023; Brenowitz et al., 2020a; Rasp, 2020; Brenowitz and Bretherton, 2018) or by fulfilling physical constraints in the network architecture (Beucler et al., 2021; Yuval et al., 2021). Careful validation setups can help the scientific community to build trust in so-called black box ML algorithms (McGovern et al., 2019).

To avoid devoting resources to the development of ML algorithms that fail in contact with reality, we encourage incorporating online testing at an early stage. ML algorithms are developed iteratively, and new versions should be tested quickly in their final place of application in the Earth system model.

The popular software libraries for ML algorithm development, such as PyTorch (Paszke et al., 2019), Keras (Chollet et al., 2023), or Tensorflow (Abadi et al., 2016), are based on the Python language. On the other hand, ICON is written in Fortran. Online testing requires either rewriting the ML emulator in Fortran, or integrating the two programming languages with one another (Brenowitz and Bretherton, 2019). Since ML algorithm development is an iterative process, frequent rewrites of the ML algorithm would be required in the former case. In order to save developer resources, we recommend coupling Python and Fortran at least during the stage of algorithm development.

In Sect. 2.1 we introduce the warm bubble scenario, which serves as a test case for SuperdropNet. The ML algorithm itself is described in 2.3. Different strategies for integrating SuperdropNet into ICON are discussed in Sect. 3. The results and the impact of SuperdropNet on atmospheric processes and prognostic variables are presented in Sect. 4.2A computational and qualitative benchmark of three different strategies is included in Sect. 4.3.

## 2  Methods

### 2.1  Warm bubble scenario

We validate SuperdropNet in the warm bubble scenario, a test case for cloud microphysics available in ICON 2.6.5. It describes an atmosphere temperature profile with a warm air bubble at the bottom that rises vertically. The test case operates on a torus grid. This grid is created by a domain of $22 \times 20$ cells where periodic boundary conditions are applied in x and y direction. The horizontal resolution is $5\,\mathrm{km}$, and there are 70 vertical levels in z direction. The simulation time step is $20\,\mathrm{s}$ with a total simulation time of $120\,\mathrm{min}$. The experiment is computationally lightweight and runs on a single compute node. We test SuperdropNet in a warm atmosphere with no ice particle formation, as well as in a mixed-phase and a cold atmosphere that both allow ice formation. All simulation parameters are summarized in Table 1. We transport the tracers required for two-moment cloud microphysics, i.e. first and second moment of the hydrometeors cloud water, cloud ice, rain, snow, graupel, and hail.

| Parameter | Description | Warm bubble | Mixed-phase bubble | Cold bubble |
|---|---|---|---|---|
| $L_D$ | Torus domain length | 5000 m | | |
| $t_{\mathrm{dyn}}$ | Dynamical time step | 20 s | | |
| $t_{\mathrm{2mom}}$ | Two-moment scheme time step | 20 s | | |
| $z_{\mathrm{lev}}$ | Atmospheric levels | 70 | | |
| $p_{\mathrm{srfc}}$ | Surface pressure | 1013.25 hPa | | |
| $T_0$ | Cold point of atmosphere | 303.15 K | 273.15 K | 268.15 K |
| $\gamma_0$ | Vertical temperature lapse rate | 0.006 K/m | | 0.009 K/m |
| $z_0$ | Altitude up to which $\gamma_0$ applies | 3000 m | | 4000 m |
| $\gamma_1$ | Lapse rate above $z_0$ | 0.00001 K/m | | 0.0001 K/m |
| $T_{\mathrm{perturb}}$ | Temperature perturbation | 10 K | | 5 K |
| $\phi_{\mathrm{bg}}$ | Background relative humidity | 0.7 | | |
| $\phi_{\mathrm{mx}}$ | Maximum relative humidity | 0.9 | | 0.95 |
| $\xi$ | Half-width of temperature perturbation in $x$ | 12500 m | | |
| $\zeta$ | Half-width of temperature perturbation in $z$ | 200 m | | 250 m |
| $x_0$ | Center of temperature perturbation in $x$ | 0 m | | |

**Table 1.** Experiment parameters for the warm bubble, mixed-phase bubble, and the cold bubble test case. Note that $t_{\mathrm{dyn}}$ and $t_{\mathrm{2mom}}$ reflect the time step used for training SuperdropNet.

## 2.2 Bulk moment scheme for cloud microphysics

In our test case, a two-moment bulk scheme is employed to compute the number concentration and total mass for all hydrometeors involved. In ICON, the bulk moment scheme used for warm rain cloud microphysics is based on Seifert and Beheng (2001). To account for collision-coalescence, the number concentration and total mass for both cloud and rain are determined by calculating the rates of collision-coalescence processes, including autoconversion, accretion, and self-collection. Here autoconversion refers to the process by which cloud droplets coalesce to form rain droplets while accretion accounts for collisions between rain and cloud droplets. Self-collection rates for cloud and rain droplets account for collisions that do not convert cloud droplets into rain. These process rates rely solely on the droplets themselves and are subsequently utilized to update the bulk moments for the following time step using a set of ordinary differential equations.

## 2.3 SuperdropNet cloud microphysics model

SuperdropNet is a machine learning emulator for superdroplet simulations in a warm rain scenario. It is a neural network consisting of fully connected layers and is trained to predict updates of the bulk moments for cloud and rain over different droplet size distributions. SuperdropNet is detailed in (Sharma and Greenberg, 2024); therefore, we will provide only a brief summary of the training procedure here.

The superdroplet simulations used for training are generated with McSnow (Brdar and Seifert, 2018). In (Brdar and Seifert, 2018) McSnow was used for simulating ice particles, while in (Seifert and Rasp, 2020) it was simulating a warm rain scenario. Similar to (Seifert and Rasp, 2020), the training data for SuperdropNet is generated in a warm rain scenario that describes only the conversion of cloud droplets into rain in a dimensionless control volume. As superdroplet simulations are stochastic in nature, we use multiple realizations of simulations to train SuperdropNet. Hence, given a set of initial conditions, SuperdropNet is completely deterministic in nature and the bulk moments estimated by it are the equivalent of averaged superdroplet simulations (Sharma and Greenberg, 2024). The microphysical processes accounted for are accretion, autoconversion and self-collection of rain and cloud droplets. In ICON, the droplet collisions corresponding to warm rain processes are treated in a separate module where the process rates for accretion, autoconversion, and self-collection of rain and cloud droplets are calculated. The parameterization scheme is localized, i.e the process rates calculated for a grid cell depend only on the rain and cloud moments corresponding to that grid cell. Other microphysical processes and the vertical transport are accounted for in separate modules, which implies that the parameterization in ICON is structured such that all individual grid points can be considered zero-dimensional boxes. Thus, the parameterization setup for droplet collisions in ICON mimics the training data for SuperdropNet. This justifies the choice of using a test scenario in ICON for online coupling and testing of SuperdropNet.

Note that only the warm rain processes are replaced with SuperdropNet. In a cold atmosphere, SuperdropNet can still be coupled to ICON, but since warm rain processes are not relevant there, including SuperdropNet is expected not to change the experiment results.

## 2.4 ICON program flow

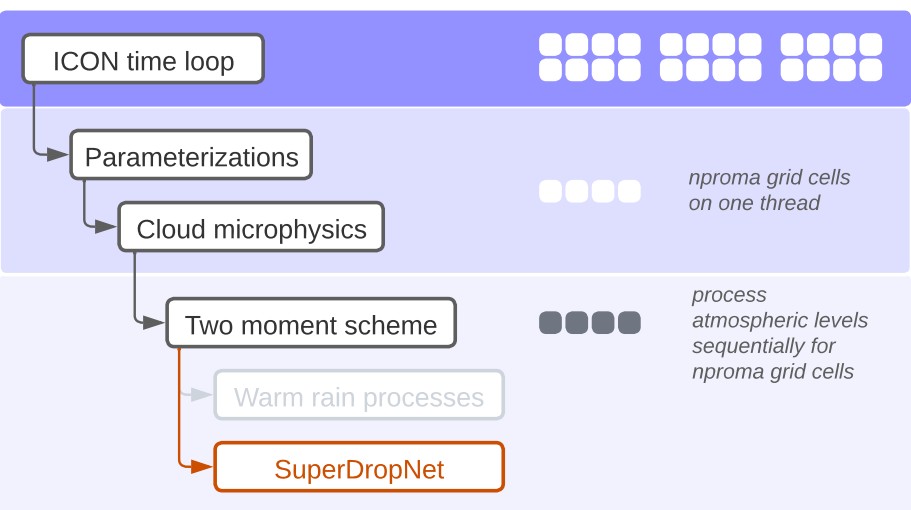

**Figure 1.** We replace the warm rain processes (gray) with a call to SuperdropNet (orange). At this point, each thread has access to an *ik-slice*, a specific representation in the cloud microphysics parameterization that corresponds to one atmospheric level for one block of grid cells.

To illustrate at which point of program execution ML-ESM coupling becomes necessary, we show the flowchart for a single ICON time step in Fig. 1, focusing only on the steps relevant to our application. Starting from the general ICON time loop, where the full grid information is available, we enter the cloud microphysics parameterization. At this point, a given thread has access to one block of grid cells with block length nproma, and all threads work in parallel. The two-moment scheme has its own grid representation, called *ik-slices*, where the block of grid cells is again divided by atmospheric levels. In our experiment, we simply replace the warm rain processes with a call to SuperdropNet, which provides updated moments for cloud and water droplets.

Since the call to the ML component is not at the grid level, but operates on *ik-slices* far down in the nested structure of the ICON program flow, we need to call SuperdropNet several times per time step – once for each block of grid cells and once for each atmospheric level. Note that saturation adjustments and evaporation are handled outside of the parts of the ICON code replaced by SuperdropNet.

## 3   Integrating SuperdropNet in ICON

There are several ways to integrate Python machine learning components into Fortran code (Partee et al., 2022). Based on a pre-selection of suitable methods, we have implemented three strategies, so-called Fortran-Python bridges. For convenience, we add a namelist to ICON that allows the selection of the coupling strategy. We perform the experiment with all three methods on the DKRZ Levante system. Levante is a BullSequana XH2000 supercomputer with 3042 compute nodes using the 3rd generation of AMD EPYC CPUs (Milan) with 128 cores per node, NVIDIA A100 GPUs, and a 130 Petabyte DDN filesystem. The nodes are connected to a Mellanox Infiniband HDR100 fabric.

### 3.1   Embedding Python as a dynamic library

Using the techniques in (Brenowitz, N., 2023), we develop a dynamic library based on Python code. The library is generated using the C Foreign Function Interface (CFFI) (Rigo and Fijalkowski, 2018) and is linked to ICON at compile time. At runtime, Python code is executed from the library. Employment of CFFI results in Python and Fortran sharing their address space, hence passing memory pointers is sufficient to access the same data. Jobs are run in a homogeneous setting, with Python code executed on the same CPU compute node as ICON.

### 3.2   Using the coupling software YAC

YetAnotherCoupler (YAC) (Hanke et al., 2023, 2016) is commonly used to couple different ICON components, e.g., atmo-sphere, ocean and I/O. YAC provides Python bindings so that external Python programs can be coupled with little effort to ICON.

YAC requires a definition of fields that are to be exchanged, and an exchange schedule that cannot be below the time step of ICON. For the warm-bubble scenario, we set the block length to the number of grid cells (880) and define two exchange fields per atmospheric level, one for the ICON-to-Python exchange, and one for the reverse exchange. This yields a total of

140 fields, that are exchanged at each time step. A smaller block length would require the developer to define more exchange fields, such that bulk moments in each grid cell can be exchanged at every time step.

     Data transfer is building on Message Passing Interface (MPI) routines that are integrated in YAC. This offers the flexibility to use heterogeneous jobs, i.e., running ICON on CPU nodes and ML inference on GPU nodes. Due to current limitations of the scheduling software employed in the DKRZ Levante system, it was not possible to schedule simulations that span the CPU

and the GPU partition of the system. Thus, we were not able to test the performance in a heterogeneous setting. With ICON shifting to GPUs, we foresee that in the future homogeneous jobs will be run on GPU nodes.

### 3.3 Pipes

We implemented a coupling between $n$ ICON processes and one Python process running on the same node using FIFO (first-in-first-out) pipes. The first ICON MPI rank on the node will spawn a separate Python process that runs a worker script. Each rank

also creates two pipes, one for each direction of communication (input and output to the Python worker). The worker iterates over all input pipes, performs the warm rain calculation on data being available and writes results back to the corresponding ICON process via its output pipe.

     While this solution does not incur the potential overhead of using MPI to communicate locally, it is not a full shared memory solution relying on pointers exclusively. The corresponding extensions to ICON and the Python worker script are optimized to

do as few memory copies as possible, though naturally some copying cannot be avoided when interacting with the pipes. As FIFO pipes only work on a local node, no cross-node setups are possible, such as running ICON and Python on different types of nodes (CPU, GPU). As the Python worker runs as a separate process on a dedicated core, the number of cores available to ICON is also marginally reduced by one.

### 3.4 Other methods

We note that the selection of methods in Sects. 3.1–3.3 is by no means encompassing all the available tools and summarize here the alternatives to the best of our knowledge:

     Four software libraries developed at ECMWF (Bonanni et al., 2022), the Cambridge Institute for Computing in Climate Science (Elafrou et al., 2023), NVIDIA (Alexeev, D., 2023), and Tongji University (Mu et al., 2023) address ML inference directly by exposing the Tensorflow and Pytorch APIs for Fortran, respectively. This adds the benefit of not requiring a Python

runtime environment at the time of execution. Since we require flexibility to use Python code beyond ML inference, and data exchange is done here via RAM comparable to the approach described in Sect. 3.1, we did not investigate these libraries further.

     During development, we noted that integrating SmartSim (Partee et al., 2022) would require a rewrite of the ICON startup routine that is beyond the scope of this project. On a similar note, the coupling routines developed in WRF-ML for the open source Weather Research and Forecasting (WRF) model cannot easily be adjusted to work with ICON (Zhong et al., 2023).

The Fortran-Keras bridge (Ott et al., 2020) allows for ML inference in Fortran based on ML algorithms developed in the Keras framework. This limits flexibility, since only those network layers and functionalities supported by the library can be used. On a similar note, the implementation of the ML algorithm in Neural Fortran (Curcic, 2019) is contingent on the library,

and the Fortran InferenceEngine (Rouson et al., 2023) is restricted to feed-forward neural networks. We chose to forego these methods since we desire the flexibility to use any novel Pytorch developments without depending on their integration into an external library.

## 4 Results

### 4.1 Experiment description

Using the three coupling techniques described in Sects. 3.1–3.3, we integrate SuperdropNet in ICON. The experiment results are the same since the same network is called, but the impact on computational performance is different. We run the warm bubble scenario and the cold bubble scenario, both with a representation of warm rain processes using SuperdropNet, as well as using the existing bulk moment scheme in the two-moment cloud microphysics module.

We compare the effect of replacing warm rain processes with SuperdropNet on the experiment outcome in Sect. 4.2. In Sect. 4.3, we compare the impact on computational performance that is incurred by integrating SuperdropNet for all three coupling techniques.

### 4.2 Comparison of the bulk moment scheme and SuperdropNet

#### 4.2.1 Rain rates

Figure 2a shows the grid-averaged rain rate in the warm bubble scenario deriving from warm rain processes using ICON's two-moment bulk cloud microphysics, with a comparison to SuperdropNet microphysics. Since SuperdropNet was trained on particle-based simulations that avoid certain statistical approximations of bulk moment schemes, we do not expect the rain rates in both scenarios to match. Due to the experimental setup, it is not possible to identify with certainty which model produces the more accurate rain rates. We do note, however, that SuperdropNet yields physically plausible rain rates. The rain rate obtained using SuperdropNet evolves in a predictable way, i.e, there is no rain at the beginning of the simulation, which eventually builds up to a peak and then slowly rescinds. At the end of the simulation, the rain rate is zero for both the simulations. No negative values are observed, and the coupling with SuperdropNet does not result in significant divergence of the simulation. This emphasizes that SuperdropNet is stable over longer simulation runs and overall behaves as a realistic ML based emulator for droplet collisions. One of the key differences in the evolution of the rain rate with the two different parameterizations is that the onset of rain is slightly delayed with SuperdropNet coupling which indicates a slower conversion of cloud droplets to rain droplets.

As a sanity check, we perform the cold bubble experiment using both the bulk moment scheme and SuperdropNet for the warm rain processes. In this scenario, warm rain processes are not relevant for the cloud microphysics, and we expect that including SuperdropNet does not affect processes with frozen particles. Figure 2b shows the grid-averaged snow rate. Both schemes show identical snow rates, which confirms that there are no undesired side-effects from coupling SuperdropNet when the conditions in the atmosphere do not allow warm-rain processes.

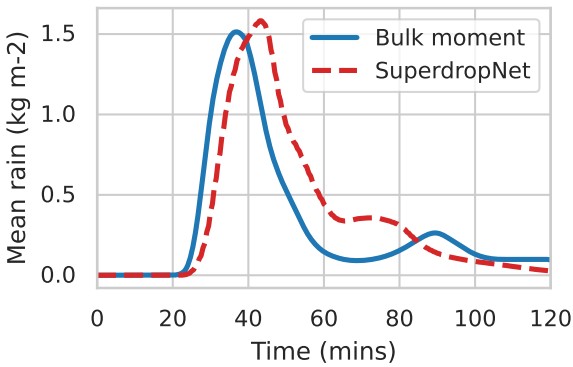

(a) Grid-averaged rain for the warm bubble scenario

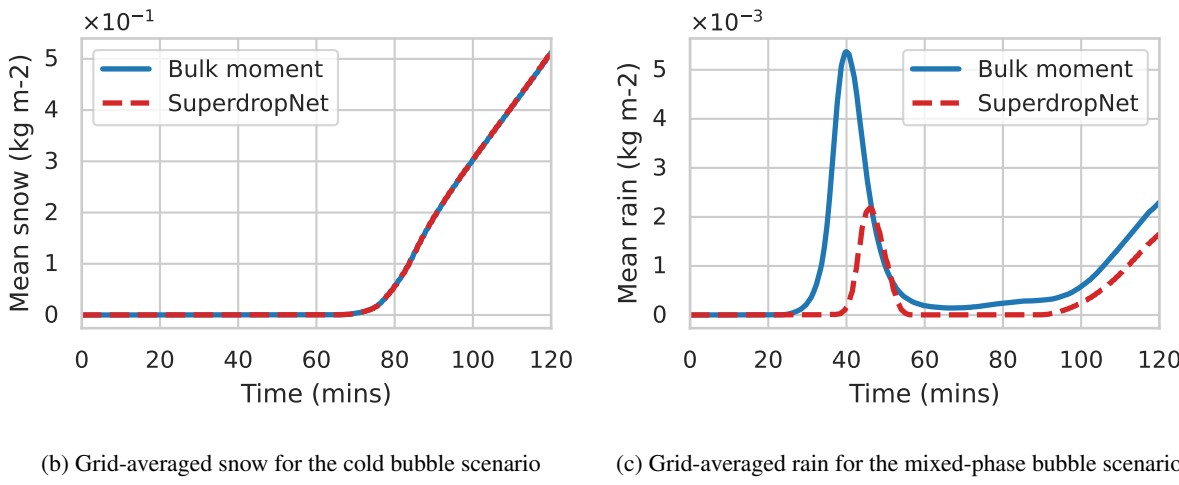

(b) Grid-averaged snow for the cold bubble scenario

(c) Grid-averaged rain for the mixed-phase bubble scenario

**Figure 2.** Grid-averaged quantities for the bulk moment scheme and SuperdropNet under various test scenarios

We also perform a mixed-phase experiment with the same setup. In this scenario, both frozen and non-frozen particles occur
in the atmosphere. Figure 2c shows the grid-averaged rain rate. The grid average values for all hydrometeors are included in
the appendix. In this case, coupling to SuperdropNet significantly drops the total rain rate. Since the total water mass remains
conserved in ICON, the suppression of rain formation leads to increased ice, cloud and snow formation (Figure A1). In ICON,
the warm rain processes are simulated before other processes such as ice nucleation, ice self-collection, snow melting etc.
Hence, SuperdropNet's effect on decreasing rain formation is subsequently reflected in the excess of other hydrometeors.

#### 4.2.2 Heat Transport Fluxes

Figure 3 shows the grid-averaged evaporative fluxes as it evolves with time during the coupled warm-bubble simulation in
ICON. While in the beginning both the bulk moment scheme and SuperdropNet produce similar fluxes, the values diverge
approximately after about 30 minutes, corresponding to the onset of rain. This difference between the magnitude of fluxes is

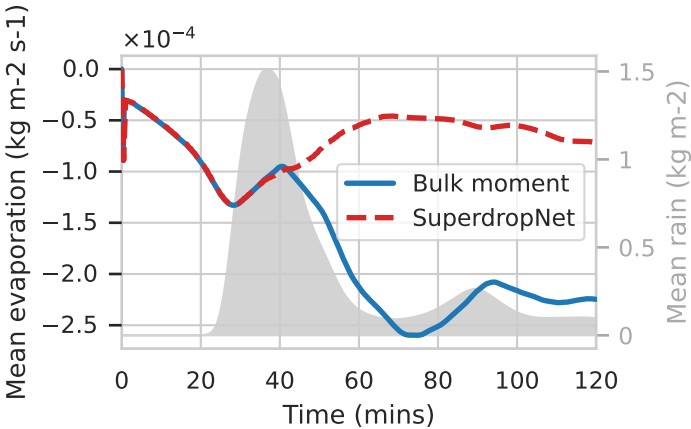

**Figure 3.** Grid-averaged evaporative heat fluxes for the bulk moment scheme used in ICON two-moment cloud microphysics, and for SuperdropNet. The gray area shows the grid-averaged rain obtained using the bulk-moment scheme (see Figure 2a). High negative values indicate a larger amount of heat transfer.

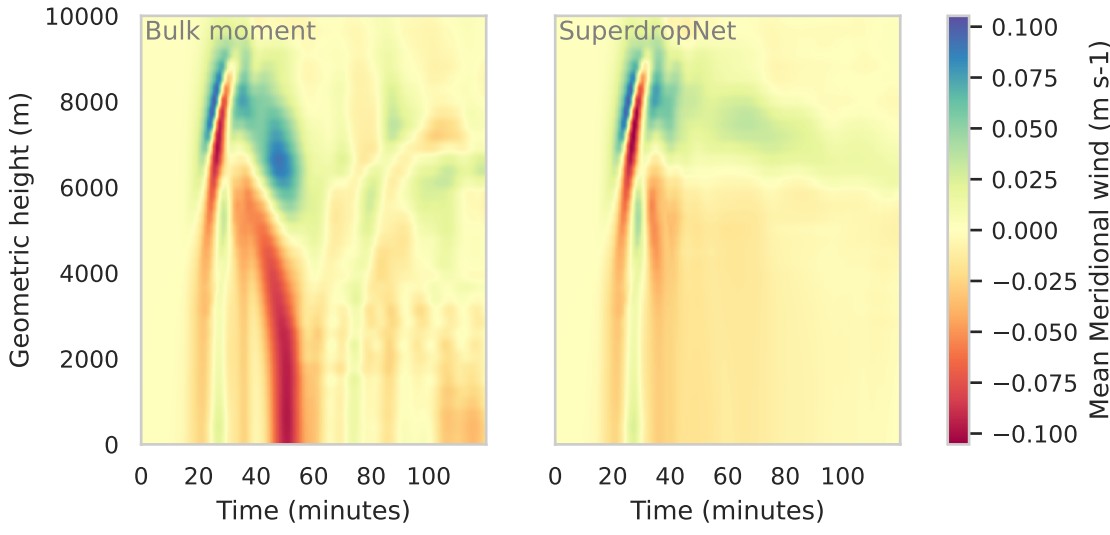

**Figure 4.** Averaged meridional winds for the bulk moment scheme used in ICON two-moment cloud microphysics *(left)* and for Superdrop-Net *(right)*.

also reflected in the evolution of winds during the simulation. Winds are the primary source of energy transport and Fig. 4 shows the evolution of meridional winds in the simulation. After approximately 40 minutes, which roughly corresponds to the end of the first rainfall with both parameterizations, the wind patterns are markedly different for the bulk moment and

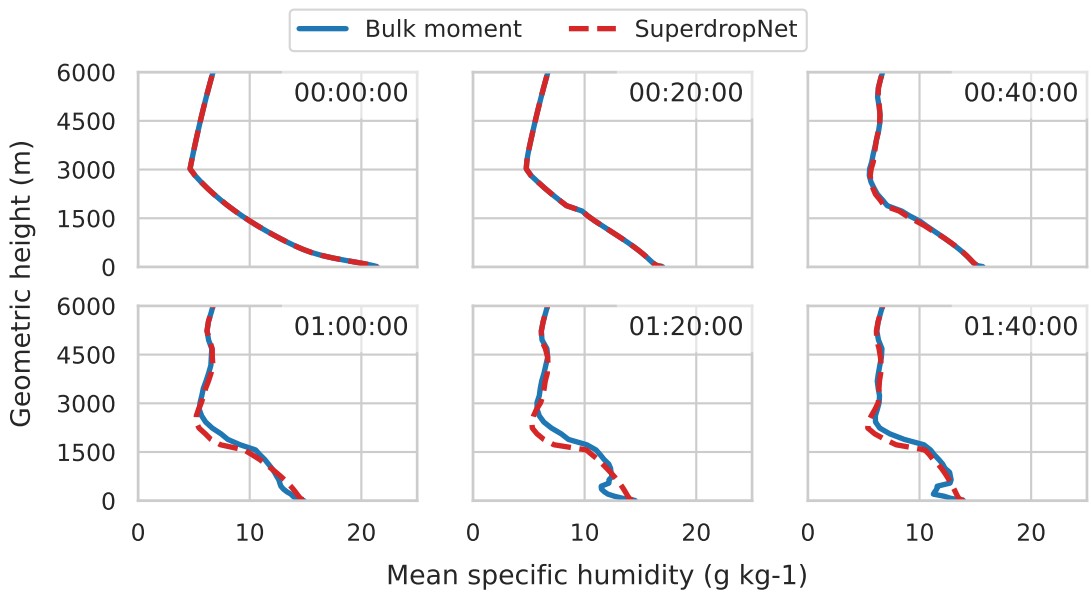

**Figure 5.** Vertical profile of the specific humidity at different times for the bulk moment scheme and for SuperdropNet.

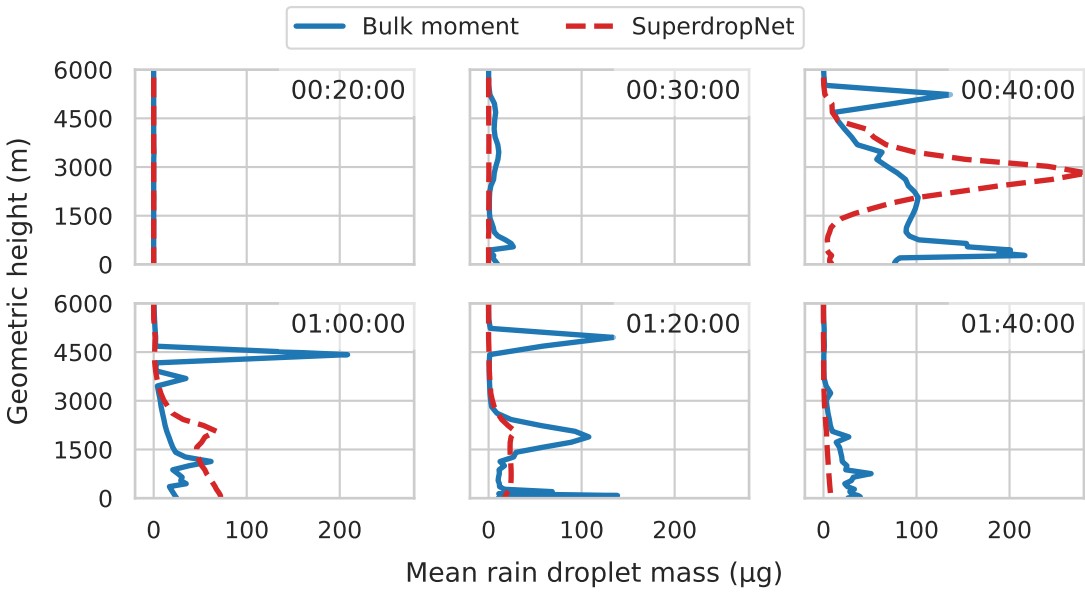

**Figure 6.** Vertical profile of the rain droplet mass, calculated as the ratio of the specific rain content and the number concentration of rain droplets at different times for the bulk moment scheme and for SuperdropNet.

the SuperdropNet parameterizations. The winds appear much stronger in case of the bulk moment parameterization across the vertical column. The reduced magnitude of winds in SuperdropNet coupling corresponds to reduced heat fluxes in Fig. 3.

Figure 5 shows the vertical profile of specific humidity at different timesteps during the simulation. For the first 40 minutes of the experiment, both parameterization schemes produce similar specific humidity profiles but this changes during the later part of the simulation. Close to the surface, it can be observed that the bulk moment parameterization produces a stronger humidity gradient in comparison to SuperdropNet. This difference in the specific humidity gradient possibly results in a higher evaporative flux for the bulk moment coupling than the SuperdropNet coupled simulation.

Similarly, in Fig. 6 the evolution of mean rain droplet mass ($\bar{X}_r$) is shown. The differences in $\bar{X}_r$ close to the surface as calculated using the bulk moment scheme vs. SuperdropNet become more visible after 40 minutes. In general with the bulk moment parameterization $\bar{X}_r$ values are higher than those with the superdroplet parameterization close to the surface. Since the evaporative flux is propotional to the mean rain mass, higher $\bar{X}_r$ in bulk moment coupling results in higher heat fluxes. Throughout the vertical column, the SuperdropNet parameterization usually corresponds to lower $\bar{X}_r$, except at the 40 minutes time step where the high $\bar{X}_r$ value near the 3000 m height also corresponds to a higher amount of the vertically integrated rain rate as seen in figure 2a.

Note that the warm-bubble scenario in ICON is highly sensitive to tiniest fluctuations within the assumptions made for cloud microphysics parameterization. Since many other complex phenomena are simplified and the focus is only on the formation and dissipation of a single cloud, small deviations in the approximation of the cloud and rain moments lead to changes in other diagnostic variables that can accumulate over time.

## 4.3 Computational performance upon including SuperdropNet

### 4.3.1 Benchmark

| Experiment | | $t_{2\text{mom}}$ (s) | Nodes |
|---|---|---|---|
| Bulk moment scheme (Fortran) | | 1.25 | 1 |
| | CFFI | 24.1 | 1 |
| SuperdropNet (Pytorch) | Pipes | 62.6 | 1 |
| | YAC | 49.5 | 2 |

**Table 2.** Time spent in the two-moment scheme in the ICON warm-bubble scenario, using the bulk-moment scheme (Fortran), and SuperdropNet (Pytorch) coupled to ICON. Note that by coupling SuperdropNet to ICON we introduce a scheme that would be computationally intractable for cloud microphysics in standard numerical simulations. A direct comparison of runtimes is therefore not possible.

We run the experiments on the Levante compute system at the German Climate Computing Center on compute nodes equipped with 2 AMD 7763 CPUs with a total of 128 cores and 256 GB main memory. The nodes are connected with a Mellanox Infiniband HDR100 fabric.

SuperdropNet provides a significant speedup by emulating processes that would otherwise be computationally infeasible to include in ICON, but when adding a Python component to the existing highly optimized Fortran code we expect an impact on computational performance. Table 2 summarizes the total time spent in the calculation of the two-moment scheme in the ICON warm bubble scenario, using the bulk moment scheme and SuperdropNet coupled to ICON through three different coupling strategies. The fastest time to solution is provided by including SuperdropNet via embedded Python, i.e. the C Foreign Function Interface (CFFI) (Sect. 3.1). Coupling SuperdropNet via YAC (Sect. 3.2) increases the relative runtime by a factor of two compared to embedded Python. Note that when coupling with YAC, the ICON and the Python main program run on two different computational nodes, which doubles the amount of computational resources required for the experiment. In the current configuration, YAC can only be used when the block length is equal to the grid size, which limits us to small experiments like the bubble scenarios. Coupling SuperdropNet and ICON using pipes is almost three times slower than embedded Python. On a qualitative note, implementing the coupling via pipes requires changes to core components of ICON beyond the cloud microphysics parameterization and may be an additional challenge for ML developers.

We note that coupling a superdroplet model directly to our test case in ICON is extremely challenging. ICON represents the warm rain processes as bulk moments, while McSnow represents them as droplet distributions. For an ideal benchmark simulation, we would need to completely overhaul the current representation of cloud microphysics processes in ICON and represent them as superdroplets for a two-way coupling. At the time of conducting this research, ICON did not allow for the representation of cloud microphysical processes as superdroplets, mainly because doing so would be computationally expensive. This is an active area of research but as of now, remains a work in progress, which makes SuperdropNet a cheaper, data-driven alternative to the superdroplet simulations.

### 4.3.2 Detailed evaluation for coupling with embedded Python

| Process | Time ($\mu$s) | Fraction |
|---|---|---|
| Time reported by ICON | $5.0 \times 10^2$ | 100% |
| Time reported by Python | $4.8 \times 10^2$ | 96% |
| $\hookrightarrow$ out of which time reported for inference | $4.4 \times 10^2$ | 87% |
| $\hookrightarrow$ out of which time reported for data transfer | $4.2 \times 10^1$ | 8.5% |

**Table 3.** Processes when coupling SuperdropNet to ICON via embedded Python and their associated duration. Machine learning inference is executed on a CPU node of the Levante compute system at the German Climate Computing Center.

We now turn to the fastest coupling scheme, embedded Python, and investigate the contribution of the individual steps to the total runtime. By including SuperdropNet, we incur computational cost for data exchange and for machine learning inference. Table 3 summarizes the contribution of the individual parts, measured with a block length of $\mathrm{nproma} = 44$ grid cells using the ICON timer module. ICON averages the execution time across a total of $496,800$ calls to SuperdropNet. Most of the time

can be attributed to model inference, while the actual data transfer is less significant. This could be attributed to the fact that ML inference has to be done on CPU. On a node equipped with an NVIDIA A100 GPU, we measure an inference time of $267\,\mu$s. This corresponds to 33% of the inference time reported on a CPU (see Table 3). Note however that a heterogeneous setup, where moments are transfered to and from the GPU nodes via the Mellanox Infiniband network, would likely lead to a larger overall wall time. Given the successful efforts of porting ICON to GPU, a future experiment could be run exclusively on GPUs. By only applying SuperdropNet when at least one input moment is nonzero, we are already reducing the number of calls to ML inference to improve performance.

## 5 Conclusions

We have coupled SuperdropNet, a machine learning algorithm emulating warm rain processes in a two-moment cloud microphysics scheme, to ICON. In the warm bubble experiment, the ML emulator is stable, and the results are physically sound.

The strategies to bridge ICON and Python provide flexibility for the development of the ML component and account for the fact that ML development is done iteratively. Both embedded Python and YAC can be integrated with little programming overhead into ICON. For a later ML emulator, that replaces a full parameterization at the grid level, YAC can be used regardless of the block length. Coupling via pipes is comparatively slow and does not scale well. Since it requires an extensive rewrite of core components of ICON, we would not recommend it for implementation. Out of the three coupling strategies we tested, embedded Python provided the fastest performance. It can be used independent of the ICON grid to execute any Python code at any level of the ICON time loop.

We note that by coupling SuperdropNet to ICON we introduce a scheme that would otherwise be computationally intractable for cloud microphysics in a standard numerical simulations. A direct comparison of runtimes is therefore not possible. Note however that integrating a Python component will slow down the overall time to solution due to the incurred cost in network inference and data transfer. For applications that are more demanding than our warm bubble scenario test case, and if the ML component is thoroughly tested, a reimplementation in Fortran would likely increase performance, at the expense of losing the flexibility of development.

A natural extension of this work are more complex modelling scenarios. This would involve training machine learning based emulators for other cloud microphysical processes and/or introduction of other hydrometeors apart from clouds and rain. Apart from droplet collisions, processes such as sedimentation of droplets and deep convection can be challenging to represent with bulk moment parameterization schemes. Hence, in the future we want to explore the possibility of creating ML based proxies for these processes while continuing to use hybrid ML-ESMs for continuous online testing.

## Appendix A: Evaluation of SuperdropNet

### A1  Mixed-phase bubble

We include the grid-averaged cloud ice, cloud water, graupel, snow, and ice for the mixed-phase experiment described in Section 4.2.1. The results are shown in Figure A1.

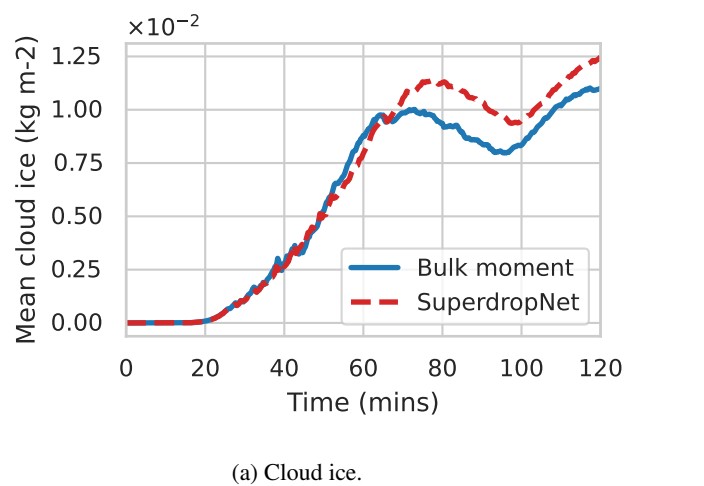

(a) Cloud ice.

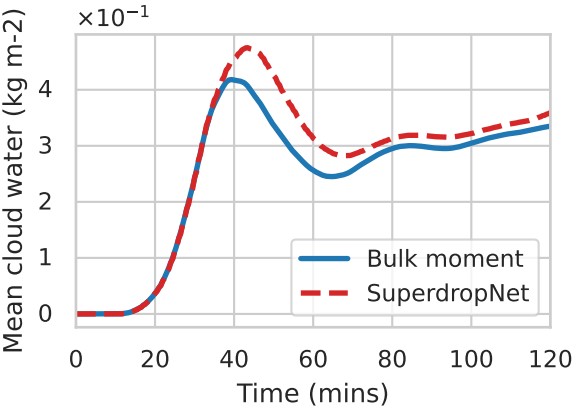

(b) Cloud water.

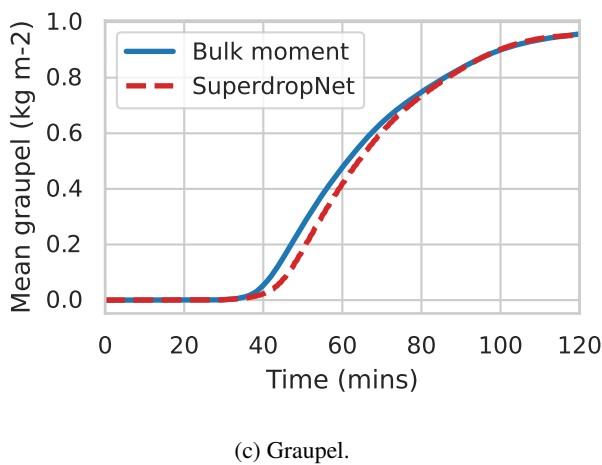

(c) Graupel.

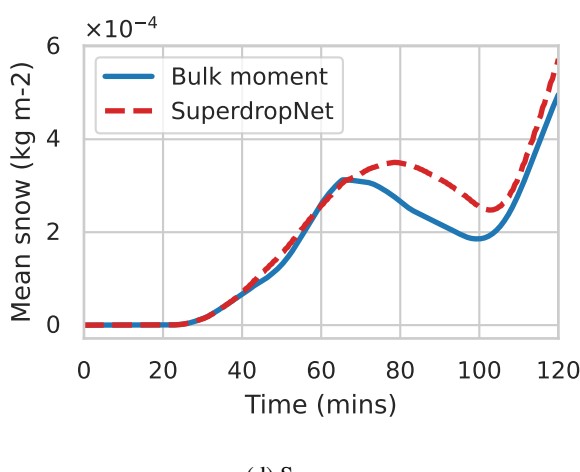

(d) Snow.

**Figure A1.** Grid-averaged rates for the mixed-phase experiment.

*Code and data availability.* The SuperdropNet (version 0.1.0) inference code, trained model weights, and modules describing the coupling between SuperdropNet inference and generic Fortran code, analysis scripts and Jupyter notebooks, as well as the experiment description files are available under the MIT license here: https://doi.org/10.5281/zenodo.10069121. The license file is included in the repository. The ICON model code used for the simulations in this paper is available under https://doi.org/10.5281/zenodo.8348256. It is based on the ICON release 2.6.5 and includes additional code for coupling SuperdropNet. ICON is now publicly available under BSD-3-C license at https://www.icon-model.org. The experiment results obtained with SuperdropNet (version 0.1.0) coupled to ICON (version 2.6.5) are available under https://doi.org/10.5281/zenodo.8348266. We used McSnow (version 1.1.0) for generating the training data in a warm rain scenario. McSnow is not publicly available. Access to McSnow can be granted upon agreeing to the ICON licensing terms by the developers of McSnow (Brdar and Seifert, 2018).

*Author contributions.* CA developed the embedded Python and YAC coupling software, performed the experiments, provided the visualizations, and led the writing of the manuscript as a whole. SS developed SuperdropNet. CA and SS defined and evaluated the experiments, curated software and data, and wrote the original draft. TW developed the pipes coupling software and contributed to the original draft. DG helped conceive the project, define the coupling task and contributed to the original draft. CA, SS, TW, and DG reviewed and edited the final manuscript.

*Competing interests.* The authors declare that they have no conflict of interest.

*Acknowledgements.* We thank A.-K. Naumann and S. Rast for support with the ICON warm bubble scenario and helpful discussions, and we thank N.-A. Dreier and M. Hanke for support with integrating YAC. This work was supported by Helmholtz Association's Initiative and Networking Fund through Helmholtz AI [grant number: ZT-I-PF-5-01]. This work used resources of the Deutsches Klimarechenzentrum (DKRZ) granted by its Scientific Steering Committee (WLA) under project ID AIM.

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
