# Peer review of "Efficient and Stable Coupling of the SuperdropNet Deep Learning-based Cloud Microphysics (v0.1.0) to the ICON Climate and Weather Model (v2.6.5)"

_EGUsphere, 2023_

## Author Comment (AC1)

We are grateful for your constructive comments on the manuscript. Below we address your comments individually. Our responses and changes are highlighted in red.

**Author's Reply to RC1:**

L73: As someone unfamiliar with ICON, I found the description of the torus grid a bit confusing. Is this a 2-dimensional domain? And it's only 5km large? Or is the resolution 5km, not the domain length?

Indeed, the resolution is 5 km, not the domain length. We changed the paragraph to read (L81):

"The test case operates on a torus grid. This grid is created by a domain of 22 × 20 cells where periodic boundary conditions are applied in x and y direction. The horizontal resolution is 5 km, and there are 70 vertical levels in z direction. The simulation time step is 20 s with a total simulation time of 120 min."

L73: Why is the timestep so short? Is this how short the dynamical timestep of ICON usually is? Is the microphysics stable with a longer timestep?

The short timestep is a reflection of the timestep on which SuperdropNet was trained. The data from the superdroplet simulations was recorded at 20s time interval. As a result, SuperdropNet can recursively predict 20s ahead.

The dynamical timestep of ICON varies with grid resolution, and the original bubble experiment used a 30s timestep. We changed that to 20s to comply with SuperdropNet's intrinsic time scale. Since SuperdropNet can predict 25 steps forward in time, the maximum permissible timestep for microphysics is 500 s(~8 minutes). Beyond which, SuperdropNet would generate erroneous predictions.

We added the following to the caption for Table 1:

"Note that $t_{dyn}$ and $t_{2mom}$ reflect the time step used for training SuperdropNet."

L98: The "cold atmosphere" comment seems a bit oversimplified. In the test later in the paper you show that there is seemingly only ice/snow without any liquid or rain. But in many situations for temperatures below freezing mixed-phase clouds are prevalent in the atmosphere. I would want to see a case with both liquid and ice to see how the ML

and bulk schemes work together, or if this presents an additional challenge to the coupling.

We thank the reviewer for pointing out the possibility of a mixed-phase scenario. We have results for the test case in a mixed-phase scenario and have added them in Fig 2c, the appendix and the discussion in Section 4.2.1, and have updated Table 1.

L182: I do not understand the phrase "neither does coupling with SuperdropNet result in dramatically different values" -- Do you mean that the 3 coupling approaches all give the same results? Or is this still referring to the first part of that sentence that says no negative values are observed?

Here we are still referring to the first part of the sentence. For clarity we changed it to(L205):

"No negative values are observed, and the coupling with SuperdropNet does not result in significant divergence of the simulation."

Fig 4: I think the latent heat flux should be proportional to the evaporation (LHF = rho*Lv*E). Therefore there is no need to show both panels (a) and (b). This is also evident because the figures look identical.

We agree that this is redundant and hence, we have removed the panel for latent heat. (Changed to Fig 3)

L202: As above, LHF and E are proportional, they both scale with the same factors. This discussion doesn't make sense to me.

We agree and have subsequently removed this discussion. We now only refer to the evaporative flux which by itself implies that similar patterns might be observed for the latent heat flux. (Section 4.2.2)

Table 3: It would be easier to read if you swap the 3rd and 4th lines of the table so that the processes are ordered in descending order of time taken. Also, to match the text, and make it easier to read, you could write the times in μs rather than s. (Like L244).

We have swapped the rows and changed the unit to μs to match with the text.

L251/260: Can you not actually couple McSnow to ICON even for this very simple dynamical warm bubble experiment? Or is this result presented in the Sharma and Greenberg (in prep) paper? It would be very nice to show a comparison between the emulator and the superdroplets in the online setting.

We agree that seeing superdroplets in ICON would be very interesting and would also

serve as a useful comparison for our results with SuperdropNet. However, the coupling of McSnow to ICON is challenging. ICON is programmed to calculate the bulk moments of the hydrometeors while McSnow, being a superdroplet based model, deals with droplet distributions directly. The two-way coupling would require completely overhauling the representation of atleast the warm rain processes such that ICON itself would have to deal with droplet distributions instead of the bulk moments. We are aware of research groups working in this direction and have been in touch with them, but as of now, this remains a work in progress. The main disadvantage of superdroplet models is their computational overhead. Even for our test cases, using a superdroplet scheme would be very expensive(even if it was feasible). This is the primary challenge that the previously mentioned research groups have been trying to address. We have added a few lines at the end of Section 4.3.1 to address this:

"We note that coupling a superdroplet model directly to our test case in ICON is extremely challenging. ICON represents the warm rain processes as bulk moments, while McSnow represents them as droplet distributions. For an ideal benchmark simulation, we would need to completely overhaul the current representation of cloud microphysics processes in ICON and represent them as superdroplets for a two-way coupling. At the time of conducting this research, ICON did not allow for the representation of cloud microphysical processes as superdroplets, primarily as doing so would be computationally expensive. This is an active area of research but as of now, remains a work in progress, which makes SuperdropNet a cheaper, data-driven alternative to the superdroplet simulations."

**Author's Reply to RC2:**

General comments:

The authors present their work coupling SuperdropNet into the ICON model. SuperdropNet is a machine learning emulator of warm rain collision-coalescence. Part of the paper explores integrating SuperdropNet into the ICON model and identifies three methods to use. The numerical performance of these methods are compared for a standard test-case of a warm rising bubble against the ICON bulk scheme.

The coupling is successful for all methods and produces reasonable results on comparison with the current bulk scheme within ICON. I found this paper of merit and an interesting read. I do have minor comments and suggestions for improvements.

I note that the emulator paper Sharma and Greenberg 2023 is in preparation, and I appreciate the authors providing in the attached assets. If possible, it would help support this work to have the article submitted before this work is published. However, it does not significantly detract from this work to not yet have Sharma and Greenberg 2023 submitted.

We have now posted a preprint for Sharma and Greenberg 2024 on arxiv and are citing that version in this study. The manuscript is under review but we hope that the preprint would suffice.

 Specific comments:

Line 25: Perhaps a more robust introduction to what an ML algorithm is, not everyone will be familiar. You could consider referencing and discuss first usages and speedups in atmospheric models, e.g. Chevallier et al 2000, Use of a neural-network-based long-wave radiative-transfer scheme in the ECMWF atmospheric model, and Krasnopolsky et al 2005, New Approach to Calculation of Atmospheric Model Physics: Accurate and Fast Neural Network Emulation of Longwave Radiation in a Climate Model. At the moment it is assumed that we all understand what an ML algorithm consists of.

We changed the paragraph (lines 20 ff) by adding an explanation of ML algorithms:

"ML algorithms are statistical algorithms that are trained on data. Neural networks are a widely used class of ML algorithms. They contain trainable parameters, the weights and biases, that are learned from data by minimizing a cost function. The trained algorithm can then be used for inference, i.e. application on unseen data of the same kind. [...]"

We added the two references in the paragraph (lines 20 ff)

"This was first established by using neural networks to emulate long-wave radiative transfer (Chevallier 2000, Krasnopolsky 2005). [...]"

Line 44: Here you say that SuperdropNet has been recently released, however I don't think SuperdropNet has yet been released. I understand that you have made your integration of SuperdropNet into ICON available. I would make the distinction here, or, point to a standalone release of SuperdropNet, as I have been unable to find it.
We agree that SuperdropNet hasn't been released so we changed the sentence(L48) to read "We present here the integration of SuperdropNet".

Line 59/60: This first sentence encourages incorporating online testing/development into an emulator. I agree with this, however it is followed by describing PyTorch, Tensorflow, and others. It almost appears that this work is implying that PyTorch etc have functionality for online testing/development. I suggest separating this encouragement of online testing/development into its own paragraph.
Line 62: States that it is necessary to integrate the two programming languages with one another. I disagree with the use of 'necessary'. Certainly I agree that it is much quicker and easier to use some sort of python-Fortran coupler, however I don't think it is entirely necessary. You could for instance write your emulator in Fortran, certainly this has been done before.
Line 63: I don't think the reference to Brenowitz and Bretherton 2019 supports the claim that it is necessary to integrate the two programming languages with each other. Brenowitz and Bretherton 2019 do provide a method of coupling, but I don't see this work supporting that it is 'necessary' to do so.
Line 64: You claim that 'hybrid ML-ESMs must be computationally powerful enough for verification experiments without requiring rewriting the ML code in Fortran'. It would certainly be helpful if you didn't have to rewrite in Fortran, if the coupled Fortran-Python code performed similar speed-wise that would be excellent news, however rewriting ML code in Fortran is a viable development route. I am aware of research groups which are doing this.
We have rephrased the paragraph(L64 onwards) to address the reviewer's four comments as follows:

"To avoid devoting resources to the development of ML algorithms that fail in contact with reality, we encourage incorporating online testing at an early stage. ML algorithms are developed iteratively, and new versions should be tested quickly in their final place of application in the Earth system model.

The popular software libraries for ML algorithm development, such as PyTorch (Paszke et al., 2019), Keras (Chollet et al., 2023), or Tensorflow (Abadi et al., 2016), are based on

the Python language. On the other hand, ICON is written in Fortran. Online testing requires either rewriting the ML emulator in Fortran, or integrating the two programming languages with one another (Brenowitz and Bretherton, 2019). Since ML algorithm development is an iterative process, frequent rewrites of the ML algorithm would be required in the former case. In order to save developer resources, we recommend coupling Python and Fortran at least during the stage of algorithm development. "

Line 73: Do you mean 2D periodic boundaries in the x-direction? The wording of 'torus grid' is perhaps less common and could be confusing.
Line 73: I assume that the top and bottom of the domain is a fixed boundary and not part of this torus? It is somewhat unclear.
Line 73: Do you mean the domain length in the x-direction is 5km? What about the z-direction?
Line 73: It would be helpful to explicitly state that this is a 2D vertical slice, which has an x and a z dimension.

We have changed the paragraph to read(L81):

"The test case operates on a torus grid. This grid is created by a domain of 22 × 20 cells where periodic boundary conditions are applied in x and y direction. The horizontal resolution is 5 km, and there are 70 vertical levels in z direction. The simulation time step is 20 s with a total simulation time of 120 min."

Line 82: What does McSnow stand for, looks like it might be a Monte-Carlo ice particle physics model. It would be worth writing this.
McSnow is introduced in Brdar and Seifert. 2018 as "a novel Monte-Carlo ice microphysics model" but this information can be misleading in our case as we use McSnow for simulating warm rain microphysics. Hence, we decided to add " McSnow, a superdroplet based cloud microphysics model" in the Introduction where McSnow is mentioned first (L50).

Line 82: The reference to Seifert and Rasp 2020 is misplaced here. In this context it reads as if Seifert and Rasp 2020 contributed to the development of McSnow, however this is not the case. I suggest separating this to stress the distinction between how Brdar and Seifert, and Seifert and Rasp, both contributed to the training of superdropNet.

We removed the reference to Seifert and Rasp 2020 (L102). We believe the next two sentences explain the role of Seifert and Rasp 2020 reference in relation to the usage of McSnow as well as presenting a simulation setup that was used for development of SuperdropNet:

"In (Brdar and Seifert, 2018) McSnow was used for simulating ice particles, while in (Seifert and Rasp, 2020) it was simulating a warm rain scenario. Similar to (Seifert and Rasp, 2020), the training data for SuperdropNet is generated in a warm rain scenario that describes only the conversion of cloud droplets into rain in a dimensionless control volume. "

Line 86: 'we use multiple realisations of simulations to train SuperdropNet' – it is my understanding that this paper has nothing to do with the actual training of SuperdropNet, and the training all happened in the (in prep) Sharma and Greenberg 2023. I would adjust this line to make the distinction obvious that Sharma and Greenberg 2003 did the training.

It is true that SuperdropNet's training happened in Sharma and Greenberg 2023. Since the stochasticity of superdroplet simulations is a major feature, describing it in this manuscript serves to give a brief account without the need for the reader to immediately turn to Sharma and Greenberg 2023.  To make the distinction clear we have added a sentence in the paragraph before being referred here, which reads(L100):

"SuperdropNet is detailed in (Sharma and Greenberg 2024); therefore, we will provide only a brief summary of the training procedure here"

Line 116: 'DKRZ Levante system'. This is the first introduction of this system, I would add a brief description of what this is.

We added the following description(L134):

"We perform the experiment with all three methods on the DKRZ Levante system. Levante is a BullSequana XH2000 supercomputer with 3042 compute nodes using the 3rd generation of AMD EPYC CPUs (Milan) with 128 cores per node, NVIDIA A100 GPUs, and a 130 Petabyte DDN filesystem. The nodes are connected to a Mellanox Infiniband HDR100 fabric."

Line 134: I would like to know what the limitations are of the DKRZ Levante system are such that you are not able to test performance in a heterogeneous setting. Perhaps

there are very good reasons, but it is unclear here.

We changed the sentence to(L154):

"Due to current limitations of the scheduling software employed in the DKRZ Levante system, it was not possible to schedule simulations that span the CPU and the GPU partition of the system. Thus, we were not able to test the performance in a heterogeneous setting. "

Lines 151-164: There are other ML libraries you could have added here. For example, there is FTA (Fortran Torch Adapter) as used in https://doi.org/10.3389/feart.2023.1149566. There is also the inference-engine written in Fortran: https://github.com/BerkeleyLab/inference-engine. I'm sure there are reasons why the authors omit these, perhaps these are not suited for this purpose, but consider adding with reasons why they are not appropriate.

We thank the reviewer for pointing us to these additional libraries for coupling Fortran and Pytorch. We note that FTA is similar to the libraries developed at ECMWF and Cambridge, and cite the reference accordingly. We also note that the InferenceEngine is restricted to feed-forward neural networks. We cite the work and discuss it accordingly (Section 3.4):

"Four software libraries developed at ECMWF (Bonanni et al., 2022), the Cambridge Institute for Computing in Climate Science (Elafrou et al., 2023), NVIDIA (Alexeev, D., 2023), and Tongji University (Mu et al., 2023) address ML inference directly by exposing the Tensorflow and Pytorch APIs for Fortran, respectively. This adds the benefit of not requiring a Python runtime environment at the time of execution. Since we require flexibility to use Python code beyond ML inference, and data exchange is done here via RAM comparable to the approach described in Sect. 3.1, we did not investigate these libraries further.

[...]

The Fortran-Keras bridge (Ott et al., 2020) allows for ML inference in Fortran based on ML algorithms developed in the Keras framework. This limits flexibility, since only those network layers and functionalities supported by the library can be used. On a similar note, the implementation of the ML algorithm in Neural Fortran (Curcic, 2019) is contingent on the library, and the Fortran InferenceEngine (Rouson et al., 2023) is restricted to feed-forward neural networks. We chose to forego these methods since

we desire the flexibility to use any novel Pytorch developments without depending on their integration into an external library."

Line 169: The bulk moment scheme in the two-moment cloud microphysics module of ICON is used, but it would be useful to include s description of how the bulk scheme functions and represents the collision-coalescence.
We have added section 2.2 that briefly describes the main approach used in the bulk moment scheme in ICON.

Figure 4: I do find the grey area a bit confusing for two reasons: 1. It does not appear to have an axis. 2. On fig4a it looks like the grey area starts at -2.5kg m-2 s-1 and on fig4b it looks like it starts at -6.5Wm-2. I think this would be rectified by adding another labelled y-axis for the grey area.
We have changed Fig 3 accordingly by adding a second y-axis to the plot. The figure caption reflects the change:

"[...] The gray area shows the grid-averaged rain obtained using the bulk-moment scheme (see Figure 1)."

Fig4 description: Heat flux is mentioned here, I would state that you have a 'larger heat flux out of the grid-cell'.
From what we understand, the evaporative and latent heat fluxes are shown as negative due to the convention prevalent in meteorology, where heat being absorbed from the surroundings for phase change is denoted as negative. To make this clear, we changed the caption for Figure 3 (numbering has changed) to "High negative values indicate a larger amount of heat transfer."

Fig7, lines 209-219: I would comment on how the SuperdropNet rain droplet profiles are very smooth in comparison to the bulk moment scheme. I would like it explained why the bulk moment scheme has these rather sharp increases at say around 4500m at 40mins, 60mins, 80mins.
Fig 6: We were unable to find any studies that imitate our exact setup to study this behavior. However, various studies have noted that lagrangian schemes have an advantage over the traditional bulk moment schemes as they prevent artificial rain formation due to the upward movement of the cloud boundary (Arabas and Shima, 2012, Andrejczuk et al., 2008, Stevens et al., 2005 ). This might explain the behavior seen in Fig 6(ordering has been changed) as the upward movement of the cloud

boundary (in case of the bulk moment scheme-blue) leads to rain formation at a higher altitude as the simulation evolves with time. But since this has not been explicitly tested in our settings, we chose not to comment directly on it in the manuscript. Most of these studies involved an LES simulation and/or observations for comparison. In our case we only have a much smaller domain in a highly idealized test scenario.

Table 2: Comparing the bulk scheme to the SuperdropNet scheme seems somewhat odd. I suppose that the scheme which SuperdropNet is based upon is not in ICON and would be difficult to put in, so I understand why this comparison was done. I would like some explanation for why this speed comparison is still relevant, and why the scheme SuperdropNet is based upon cannot be put into ICON directly. I think you mention this in your conclusions, but a brief sentence here would be helpful too.
We believe that adding the timing comparison in Table 2 shows that adding an externally coupled module leads to a significant slow down. We changed the caption of Table 2 to read:

"Time spent in the two-moment scheme in the ICON warm-bubble scenario, using the bulk-moment scheme (Fortran), and SuperdropNet (Pytorch) coupled to ICON. Note that by coupling SuperdropNet to ICON we introduce a scheme that would be computationally intractable for cloud microphysics in standard numerical simulations. A direct comparison of runtimes is therefore not possible."

We have added the following lines at the end of section 4.3.1(L263) explaining why McSnow cannot be coupled to ICON as of now:

"We note that coupling a superdroplet model directly to our test case in ICON is extremely challenging. ICON represents the warm rain processes as bulk moments, while McSnow represents them as droplet distributions. For an ideal benchmark simulation, we would need to completely overhaul the current representation of cloud microphysics processes in ICON and represent them as superdroplets for a two-way coupling. At the time of conducting this research, ICON did not allow for the representation of cloud microphysical processes as superdroplets, primarily as doing so would be computationally expensive. This is an active area of research but as of now, remains a work in progress, which makes SuperdropNet a cheaper, data-driven alternative to the superdroplet simulations."

Line 225: Perhaps I have missed it in this work, but how much is the speedup of SuperdropNet over the model it is based on? This seems like an important thing to include.

From our experience of generating training data from McSnow, and running offline inference on SuperdropNet, the speedup varies. The speedup depends entirely on the initial conditions of the simulation that are assumed.  For simulations using a small amount of initial water content, SuperdropNet speeds up the simulation by atleast a factor of 100 whereas for simulations which start with a sufficiently high quantity of water, the speedup might only be by a factor of 2. The primary reason for this is that SuperdropNet's inference time per simulated minute is constant whereas for McSnow, it scales with the initial conditions of the simulation. Since this information pertains to running SuperdropNet offline and comparing it to the training simulations from McSnow, we decided against adding this information in this manuscript as here we focus on "online" coupling only.

Line 263: I would argue that it 'would likely' rather than 'might' increase performance. It's fairly clear that you would see speedups by fully integrating a superdropNet model written in Fortran within ICON. I do agree however, that losing flexibility of development is a good reason why this was not done.

We agree and have changed the wording from "might" to "would likely".(L296)

Technical corrections:

Line 7: ICON acronym should be properly introduced here.

We added the complete name in line 7

Line 22: Reference order should be in date order, i.e. Christensen and Zanna 2022 should come first.

We rearranged the order of references here to match with the rest of the manuscript(newest first).

Line 29: Brenowitz and Bretherton (2018) should be in parenthesis.

Added the parenthesis

Line 41: 'total droplet concentration and the total liquid water content' would be more explicit.

We changed it to "total droplet concentration and the total water content"(L45)

Line 55/56: References should be written in date order.

Rectified the order (L59)

Line 71: 'testcase' should read 'test case'.

Changed to "test case" (L74)

Line 119: The reference to 'Rigo and Fijalkowsi (2018)' should be within parenthesis to read better.

Added parenthesis (L140)

Line 171: 'incurred' not 'incured'.

Changed to "incurred" (L194)

Fig4 description: Replace 'higher' with 'larger'. Higher can be ambiguous and be interpreted as 'more positive'.

Changed to "larger" (Fig 3)

Line 263: 'loosing' should be written 'losing'.

Changed to "losing" (L296)

---

## Author Response (AR2)

We are grateful for your constructive comments on the manuscript. Below we address your comments individually. Our responses and changes are highlighted in red.

**Reply to the Editor:**

Please address the technical corrections pointed out by the Reviewer:

- L93: "did not" should be "do not"
- L221: "evaporative fluxes" missing "s"

THese have been corrected

Please also note that the https://code.mpimet.mpg.de/projects/icon-license webpage linked from "Code and data availability"
section no longer exists (403 error: "The project you're trying to access has been archived"). Please update the link and licensing information accordingly.
We have updated the link and added information about ICON's changed licensing terms.

In references, please:

- add DOI for Chevallier et al. 2000: https://doi.org/10.1002/qj.49712656318
- add DOI for Krasnopolsky et al. 2005: https://doi.org/10.1175/MWR2923.1

We have added the missing DOIs.